# Development of New Restorer Lines Carrying Some Restoring Fertility Genes with Flowering, Yield and Grains Quality Characteristics in Rice (*Oryza sativa* L.)

**DOI:** 10.3390/genes13030458

**Published:** 2022-03-03

**Authors:** Mamdouh M. A. Awad-Allah, Nehal M. Elekhtyar, Mohamed Abd-El-Moaty El-Abd, Mohamed F. M. Abdelkader, Mohamed H. Mahmoud, Azza H. Mohamed, Mohamed Z. El-Diasty, Manal M. Said, Sahar A. M. Shamseldin, Mohamed A. Abdein

**Affiliations:** 1Rice Research Department, Field Crops Research Institute, Agricultural Research Center, Giza 12619, Egypt; momduhm@yahoo.com (M.M.A.A.-A.); nehal_rrtc@yahoo.com (N.M.E.); 2Agronomy Department, Faculty of Agriculture, Kafr El-Sheikh University, Kafr El-Sheikh 33516, Egypt; mohamedel3bd@yahoo.com; 3Department of Plant Production, College of Food and Agriculture, King Saud University, Riyadh 12372, Saudi Arabia; mohabdelkader@ksu.edu.sa; 4Department of Biochemistry, College of Science, King Saud University, Riyadh 11451, Saudi Arabia; mmahmoud2@ksu.edu.sa; 5Citrus Research & Education Center, University of Florida, IFAS, 700 Experiment Station Road, Lake Alfred, FL 33850, USA; azza@ufl.edu; 6Genetic Department, Faculty of Agriculture, Mansoura University, Mansoura 35516, Egypt; m_z_diasty@mans.edu.eg; 7College of Biotechnology, Misr University for Science and Technology, Giza 12563, Egypt; manalfaris@yahoo.com; 8Botany Department, Women’s College for Arts, Science and Education, Ain Shams University, Cairo 11566, Egypt; shams.sahar@women.asu.edu.eg; 9Seed Development Department, Agricultural Professions Syndicate, Downtown, Cairo 11669, Egypt

**Keywords:** rice, grain yield, yield components, floral traits, genetic variability, coefficient of variation, heritability, genetic advance

## Abstract

This study was carried out using 22 promising restorer lines of rice and their parental lines to study genetic variability and genetic advance for yield and yield-associated grain quality traits and floral traits. These genotypes are evaluated in a replicated trial using Randomized Complete Block Design (RCBD) with three replications at the Experimental Farm of Sakha Agricultural Research Station, Sakha, Kafr El-Sheikh, Egypt, during the seasons from 2012 to 2020. Analysis of variance revealed that highly significant variations were observed among the genotypes for all the studied characters. Both GCV% and PCV% were high for the number of spikelets per panicle, the number of filled grains per panicle, and panicle weight. The genetic advance in the percentage of mean was high for days to plant height, panicle length, number of spikelets per panicle, number of filled grains per panicle, panicle weight, grain yield per plant, anther length, anther breadth, duration of floret opening, and head rice percentage. Mean performance of the rice genotypes indicated that the genotypes NRL 59, NRL 55, NRL 62, NRL 63, NRL 66, and NRL 54-2 were promising for grain yield and associated desirable traits. Thus, some of these promising lines can be promoted as a new rice variety and could be used as a source for developing new hybrid combinations in hybrid rice breeding programs. The percentage of advantage over better parent and Giza 178 as the commercial variety was significant and there were highly significant desirable values among the genotypes for all the studied traits in the two years, indicating that the selection is effective in the genetic improvements for these traits.

## 1. Introduction

Rice is a staple food and feed crop. Rice is one of the most important crops for almost all of the world especially developing countries [1,2]. Most of the traits of interest to breeders are complex and are the result of the interaction of several components [3]. Cultivation of hybrid rice leads to a significant improvement in rice yield. Generally, the grain yield is increased by more than 20% compared to inbred rice varieties [4]. The cytoplasmic male sterility (CMS) is associated with a mitochondrial mutation that causes an inability to produce fertile pollen. The fertility of CMS plants is restored in the presence of a nuclear-encoded fertility restorer (*Rf*) gene. This is found in lead rice-type CMS, discovered in the indica variety ‘Lead Rice;’ the fertility of the CMS plant is restored by the single nuclear-encoded gene *Rf2* in a gametophytic manner. Researchers have performed the map-based cloning of *Rf2*, and proved that it encodes a protein consisting of 152 amino acids with a glycine-rich domain. The expression of *Rf2* mRNA was detected in developing and mature anthers. An *RF2*–GFP fusion was shown to be targeted to mitochondria. Replacement of isoleucine by threonine at amino acid 78 of the *RF2* protein was considered to be the cause of functional loss in the *rf2* allele. As *Rf2* does not encode a pentatricopeptide repeat protein, unlike a majority of previously identified *Rf* genes, the data from this study provide new insights into the mechanism for restoring fertility in CMS [5]. CMS is a maternally inherited trait, and is often associated with the unusual ORFs found in mitochondrial genomes [6]. The nuclear-encoded genes known as fertility restorer genes (*Rf*) leading to pollen fertility can be restored in some CMS lines. The hybridization of a CMS line, a maintainer line, and a restorer line carrying the restorer gene is useful in the development of hybrid varieties. In addition to their commercial use, CMS/*Rf* systems have attracted much interest for their role in elucidating genetic interactions and the cooperative functions of mitochondrial and nuclear genomes in plants [7].

The system of (CMS) is commonly used in hybrid rice breeding. The wild abortive (WA) is most important of three main CMS sources (WA, BT, and HL), that are popularly used in commercial hybrid rice seed production. The location of the restorer genes for CMS lines with DNA markers has been investigated during the last few years. The restorer gene for the BT-type CMS, *Rf-1*, has been mapped to chromosome 10 [8,9]. However, reports regarding the locations of restorer genes for the WA-type CMS are inconsistent. Several chromosomal locations have been reported: two on chromosome 1 [10,11,12], one on chromosome 7 [13], two on chromosome 10 [11,13,14], and four QTLs on chromosomes 2, 3, 4, and 5 [15]. The restorer gene for the HL-type CMS designated as *Rf-5* was located on chromosome 10 [16].

In Egypt there is only one commercial restorer line of rice, i.e., Giza 178. However, despite its distinction in many characteristics, as a restorer line, it has some defects, such as the short (lowest) plant height compared with CMS lines, which leads to the lowest out-crossing in the field of seed production. There are CMS lines early in flowering, moreover the other CMS lines will be needed because it has perfect synchronization with the other restorers. Giza 178 is also disadvantaged in having a low weight for a thousand grains, and the small size of the grain (grain shape), in addition to the fact that the hybrid rice breeder needs different restoring fertility lines with higher a heterotic effect and general combining ability effects, [17,18].

Despite tremendous success, cytoplasmic male sterility (CMS) systems suffer from several fundamental problems. The main problem is that *Rf* genes are found in only 2 to 5% of rice germplasm; therefore, only a very small number of rice germplasms can be explored as restorer lines for heterosis [19], which make it difficult to breed superior hybrids [20]. Besides this, it is also tedious to breed new restorer lines. Because the restorer line needs to carry the very specific *Rf* genes for the CMS genes, breeders not only need to select for the desired traits for plant development and stress tolerance, etc., but also need to make sure that the *Rf* genes and combining ability are not lost from generation to generation, and this significantly increases the workload and difficulties during breeding. However, few genotypes show strong restoration ability as effective restorers of cytoplasmic male sterility (CMS) in the development of hybrid rice [21]. To meet the expanding horizons of hybrid rice improvement, it is essential to assemble, evaluate, improve, and conserve the parental lines [22]. The development of iso-cytoplasmic restorer lines from elite rice hybrids through a process of generation advancement followed by selection is a novel approach for the development of new restorer lines. Fertility restoration in rice CMS systems is facilitated by nuclear–cytoplasmic interactions that can display varying fertility levels. Iso-cytoplasmic lines are those carrying the same cytoplasm in which interactions between the cytoplasm and the fertility restoring (*Rf*) genes are maintained at a similar level. Iso-cytoplasmic restorer lines derived from wild abortive WA-CMS based rice hybrids with more than 85% spikelet and pollen fertility [23] have a unique feature, in that they all carry WA cytoplasm [24]. An effective male sterility system is used to produce hybrid seeds in large quantities in hybrid rice seed production. In hybrid rice breeding, the pollen grains of the cytoplasmic male sterile lines (CMS) are sterile and the female organ of the CMS lines depends on the fertile pollen released from maintenance or restorer lines to produce seeds. The floral trait of parental lines, i.e., CMS lines, its corresponding maintainer, and restorer lines are the main factors in hybrid rice seed production because they influenced outcrossing between parental lines [25].

The cultivation of hybrid rice is very limited, due to the high cost of seeds and the low production capacity of hybrid seeds. To overcome and avoid these challenges, it is necessary to increase the yield of hybrid seeds by improving the outcrossing percentage of crosses between parental lines in hybrid rice seed production [26]. 

The characterization of the parental lines of floral traits is an essential aspect of every breeder and this should be kept in mind while selecting the parental lines in a hybrid rice breeding program. The extent of outcrossing on male cytoplasmic sterile lines (CMSs) is influenced by their floral traits, that is, the length and width of the anther as well as the length of the filaments, in the pollen parent [27]. 

The information about certain genetic parameters of variability for different traits of economic significance is important for plant breeders before releasing any variety. The presence and magnitude of genetic variability in a gene pool is the prerequisite of a breeding program. Heritability estimates provide information on the proportion of variation that is transmissible to the progenies in subsequent generations. Knowledge of heritability plays a major role in selection-based improvement of the crop because it indicates the extent to which traits can be passed on to future generations. Therefore, genetic variability is the prerequisite for making progress in crop breeding programs [28]. Genetic advances provide information on the expected genetic gain resulting from the selection of superior individuals. Variance is an essential factor that determines the amount of progress to be expected from the selection. As the phenotypic variance does not directly show its effectiveness in selection for obtaining genetic improvement unless the genetic part of the variance is known. Hence, insight into the magnitude of genetic variation available is of paramount importance for plant breeders to initiate a successful breeding program. It becomes necessary to partition phenotypic variance into heritable and non-heritable components with the help of genetic parameters such as genotype and phenotype co-competence, heritability, and genetic advance to facilitate the selection. Pedigree selection is one of the oldest and most widely used breeding methods in rice improvement. This method is highly appropriate for developing rice. One of the major advantages of pedigree selection is that it involves the combination of many genes [29].

The present investigation was designed to develop new iso-cytoplasmic restorer lines from a set of six different promising rice hybrids developed in Egypt; these hybrids were identified as being of the constituent restorer fertility lines, carrying the Rf1, Rf3, and Rf4 restorer fertility genes, and the evaluation thereof to identify effective iso-cytoplasmic rice restorer lines which can further be utilized in the development of improved rice hybrids. In addition, the analytical characterization and estimation of genetic variability, heritability, and genetic advance of grain yield, yield-contributing traits, and grain quality in 29 rice genotypes is undertaken.

## 2. Materials and Methods

### 2.1. Genetic Materials

The experimental materials for this study consisted of 252 iso-cytoplasmic restorer lines derived from promising rice hybrids in Egypt and evaluated for various grain yield, yield-contributing traits, and some grain quality traits. Growing the F_1_ plants and selfing to produce the F_2_ generation was done in the nursery. The selection started in 2013 as F_2_ and continued up to F_7_ in 2019. Promising lines were selected and grown along with the parental lines and evaluated for phenotypic performance and yielding ability during the 2019 and 2020 rice seasons. A total of 22 lines (Table 1) of their performance according to their phenotypic, grain yield, yield acceptability, and grain quality were selected for evaluation in randomized complete block design with the Giza 178 rice variety (the only commercial restorer line) as a check in three replications during the 2019 and 2020 rice seasons.

Each entry was grown in 5 rows within each block in a randomized complete block design (RCBD) with three replications. Each replicate consisted of 29 genotypes randomized and replicated within each block. Twenty-five-day-old seedlings were transplanted 20 cm apart between rows and 20 cm within the row. The complete recommended package of cultural practices was applied.

### 2.2. Field Evaluation

Five plants were randomly selected from the central rows in each replicate and evaluated for grain yield, its components, floral traits, and some grain quality traits.

Data were collected on no. of days to 50% heading (day), plant height (cm), number of panicles per plant, panicle length (cm), number of spikelets per panicle, spikelet fertility (%), number of filled grains per panicle, panicle weight (g), 1000 grain weight (g), grain yield per plant (g), anther length (mm), anther breadth (mm), filament length (mm), duration of floret opening (min), hulling percentage, milling percentage, and head rice recovery percentage. 

All the measurement characters were done according to IRRI Standard Evaluation System for rice [30]. 

Hulling and milling %:

The percentage of hulling and milling recovery was calculated according to the formula found in [31] as follows:Brown rice (%) = (weight of brown rice/weight of rough rice) × 100
Total milled rice (%) = (weight of total milled rice/ weight of rough rice) × 100

### 2.3. Statistical Analysis

#### 2.3.1. Estimation of Genetic Components

The data were statistically analyzed using the ANOVA analysis of variance based on the model proposed by [32]. 

The magnitude of the components of variances has been obtained from the analysis of variance to appraise the different genetic parameters as described by [33,34]. 

The genotypic and phenotypic variances were calculated as per the formulae proposed by [35]. The genotypic (GCV%) and phenotypic (PCV%) coefficient of variation was calculated by the formulae given by [35]. Heritability in a broad sense [h^2^_(bs)_] was calculated by the formula given by [36] as suggested by [37]. From the heritability estimates, the genetic advance (GA) was estimated by the following formula given by [37]. 

Mean squares were used to estimate:σ^2^_g_ = (MSS − MSE)/r

σ^2^_ph_ = σ^2^e + σ^2^_g_, where broad-sense heritability (h^2^_bs_) was estimated as follows:

h^2^_bs_ = (σ^2^_g_/σ^2^_ph_) × 100 and the phenotypic (PCV) and genotypic (GCV) coefficients of variation were computed as follows:PCV = 100 × √σ^2^_ph_/X^−^
GCV = 100 × √σ^2^_g_/X^−^
GA = k × h^2^_bs_ × √σ^2^_ph_

Expected genetic advance (GA): Expected genetic advance from direct selection for all studied traits was calculated according to [33] as follows:GA% at 5% (selection intensity) = 100 × k × h^2^_bs_ × σ^2^_ph_/X‾ or: GA% = (GA/X‾) × 100
where X‾: general mean and k is selection differential (k = 2.06 for 5% selection).

Ref. [38] categorized the value of GCV and PCV as: low = 0–10%

Moderate = 10–20% and high = >20%.

As suggested by [37], h^2^_bs_ estimates were categorized as:Low = 0–30%; medium = 30–60% and high = above 60%

#### 2.3.2. Estimation of the Advantage over Better Parent and Commercial Variety

##### The Advantage over Better Parent

The advantage over better parent was calculated as percentage of the newly developed restorer lines over its better parent (BP) [39].

$$\mathrm{The}\;\mathrm{advantage}\;\mathrm{over}\;\mathrm{better}\;\mathrm{parent}\;(\mathrm{ABP})=\frac{\bar{M}-B.P}{B.P}\times 100$$


##### The Advantage over Commercial Variety

The advantage over the high yielding commercial variety calculated as percentage of increased or decreased of the newly restorer lines over the commercial one (CK) [39].

$$\mathrm{The}\;\mathrm{advantage}\;\mathrm{over}\;\mathrm{commercial}\;\mathrm{variety}\;(\mathrm{ACK})=\frac{\bar{M}-CK}{CK}\times 100$$


Appropriate LSD values were calculated to test the significance of the advantage over commercial variety, and the advantage over better-parent, according the method suggested by [40]:
$$\mathrm{LSD}\;\mathrm{for}\;(\mathrm{ABP}\;\mathrm{and}\;\mathrm{ACK})=t_{0.01}^{0.05}\;\sqrt{\frac{2\mathrm{MSe}}{r}}$$

where:

*t*: Value at certain probability level and given degrees of freedom for error.

MSe: Error mean squares from the analysis of variance. 

r: Number of replications.


$\bar{M}$
: The mean of the newly developed restorer lines for a character. 

*B.P*: The better parent mean in the newly developed restorer lines.

## 3. Results

### 3.1. Developing New Iso-Cytoplasmic Restorers Lines 

A set of 252 iso-cytoplasmic restorer lines of rice were developed from six promising rice hybrids in Egypt. Out of 252 is cytoplasmic restorer lines, 94 lines based on their phenotypic and yield acceptability were selected for crosses with two cytoplasmic male sterile lines (Figure 1). Out of 94 cytoplasmic restorer lines, 22 lines were selected based on the results of the test cross and their phenotypic and yield acceptability to evaluate, along with seven parental lines, including a check variety, for various grain yield, yield-contributing traits, floral traits, and grain quality traits. The comparison of different iso-cytoplasmic restorer lines was performed in three replications with a check variety. 

### 3.2. Test Cross Experiment

Pollen fertility and spikelet fertility for hybrid combinations as shown in (Appendix A). The pollen fertility percentage of tested hybrids varied from 89.23% (IR69625A × NRL 65) to 98.38% (G 46A × NRL 64). 

In contrast, the spikelet fertility percentage of tested hybrids varied from 87.08% (IR69625A × NRL 65) to 95.12% (G 46A × NRL 62).

### 3.3. Mean Performance

The newly developed selected restorer lines have better performance in all studied traits (Appendix A). For the days to 50% heading, the data showed that the new developed restorer lines NRL 52 and NRL 53 were early in two rice seasons, and the time to 50% heading was 68.3, 68, 75.23, and 76.23 days, respectively. Concerning plant height, the five newly developed restorer lines, NRL 70, NRL 73, NRL 72, NRL 68, and NRL 71, were short in plant height and showed the lowest mean values in the first and the second season, while the rice genotypes NRL 55, NRL 58, NRL 59, NRL 54-2, and NRL 65 showed the highest values of plant height in the first season and the second season. For the mean values of the number of panicles per plant, the genotypes NRL 74, NRL 37, NRL 62, and NRL 53 showed the highest mean values in the first season and the second season, respectively. Concerning panicle length; the genotypes NRL 37, NRL 62, NRL 55, NRL 54-2, NRL 46, NRL 58, and NRL 59 gave the highest panicle length in the two seasons, while the check variety Giza 178 had a shorter panicle length in the two seasons. Concerning the number of spikelets per panicle, the highest value was recorded for the lines NRL 37 over the two seasons. 

The genotypes NRL 52, NRL 72, NRL 67, and NRL 74 showed highly spikelet fertility (Appendix A). Concerning the number of filled grains per panicle, the newly developed restorer lines NRL 37 and NRL 26 showed the highest values in the first and the second year. For mean values of panicle weight, the lines NRL 54-2, NRL 37, and NRL 58 gave the highest values of at the second and first year. Concerning the 1000-grain weight, the genotypes NRL 55, NRL 46, NRL 52, NRL 72, and NRL 54-2 showed the highest values in the first year and the second year (Appendix A and Figure 2). Regarding the grain yield/plant, the genotypes NRL 59, NRL 55, NRL 62, NRL 63, and NRL 66 showed the highest mean values in the first year and the second year, while the check rice variety Giza 178 gave the lowest value of the grain yield/plant at the second and first season (Appendix A and Figure 3). 

For the mean performances of floral traits studied, i.e., anther length (mm), anther breadth (mm), filament length (mm), and duration of floret opening (min) of all the studied genotypes Appendix A, the newly developed restorer line NRL 54-2 showed the highest values in the first and second season of these traits (Figure 4A,B).

For hulling percentage, the genotype NRL 52 showed the highest mean values in the first and second seasons. Concerning milling percentage, the genotype NRL 55 showed the highest mean values in the two years. For head rice percentage, the rice genotypes NRL 46 and NRL 59 recorded the highest mean values in the second and first season (Appendix A and Figure 5).

All developed lines were resistant to both blast disease and the rice-stem borer insect (Appendix A).

### 3.4. Analysis of Variance 

Analysis of variance and CV% are shown in Table 2; the results show that there are highly significant differences between all genotypes for all the traits studied.

### 3.5. Phenotypic, Genotypic Coefficient of Variation, and Genetic Advance

The genotypic and phenotypic coefficient of variation in different traits maintained correspondence for all the traits under study. In general, the phenotypic coefficient of variation was higher than the genotypic coefficient of variation. 

The results revealed a high estimate of phenotypic and genotypic coefficient of variation for spikelets per panicle, number of filled grains per panicle, and panicle weight (g) (Table 3). A moderate value of the phenotypic and genotypic coefficient of variation was observed for plant height, the number of panicles per plant, panicle length, grain yield/plant, anther length, anther breadth, filament length, duration of floret opening, and head rice percentage (Table 3), while lower values of the phenotypic and genotypic coefficient of variation were observed for days to 50% of heading, spikelet fertility percentage, 1000-grain weight (g), hulling percentage, and milling percentage (Table 3). 

The genetic advance in percentage (expected) of the mean was high for plant height, number of panicles per plant, number of spikelets per panicle, number of filled grains per panicle, panicle weight, grain yield per plant, anther length, anther breadth, duration of floret opening, and head rice percentage (Table 3). Moreover, the genetic advance in percentage (expected) of the mean was moderate for days to 50% of heading, panicle length, 1000-grain weight (g), and filament length, while low genetic advances in percentage (expected) of the mean were observed for the spikelet fertility percentage, hulling percentage, and milling percentage. High heritability with low genetic advance was observed for the spikelet fertility percentage, hulling percentage, and milling percentage (Table 3).

### 3.6. Estimation of Heritability

In this study, the traits expressed moderate to high heritability estimates ranging from 66.19 to 99.41 and 48.84 to 99.40 percent in the first and second years, respectively. High heritability was observed for the traits days to 50% heading, plant height, panicle length, number of spikelets per panicle, number of filled grains per panicle, panicle weight, 1000-grain weight, grain yield per plant, anther length, anther breadth, filament length, duration of floret opening, and head rice percentage (Table 3). Moderate heritability was observed in the number of panicles per plant (56.07%) in spikelet fertility percentage (49.39%), hulling percentage (58.58%), and milling percentage (48.84%) in the second year.

### 3.7. The Advantage over Better Parent and Commercial Variety

The data in Appendix A show that the percentage of advantage over better parent and Giza 178 as the commercial variety was significant and highly significant among the genotypes for all the studied traits in the two years. For days to 50% heading and plant height, many lines showed significant and highly significant negative estimates over better parent as well as over Giza 178 as a commercial variety. While 15 new lines showed significant and highly significant positive estimate value advantages over Giza 178 as a commercial variety; see Appendix A and Figure 6. 

Regarding number of panicles/plant, 13 newly developed restorer lines showed significant and highly significant estimates values in both years. Concerning panicle length, 13 new restore lines gave significant or highly estimated values in the first and second years. For the number of spikelets per panicle, 12 newly developed restorer lines showed significant and high estimate values in the first and second year over better parent. The results revealed that 12 lines showed significant and highly significant positive estimate advantages over Giza 178 as a commercial restorer line; see Appendix A and Figure 7. 

For panicle weight, the newly developed restorer lines showed significant or highly significant positive estimate advantages over Giza 178 as a commercial restorer line; see Appendix A and Figure 8. Concerning 1000-grain weight, most of the lines showed significant and highly significant positive estimate advantages over better parent, while all new restorer lines showed highly significant positive estimate advantages over Giza 178 as a commercial variety (Appendix A) and Figure 9.

Regarding grain yield, the data showed that most of the lines (19 lines) showed significant and highly significant positive estimate advantages over better parent as well over Giza 178 as a commercial variety (Appendix A and Figure 10). Most of the newly developed restorer lines showed significant and highly significant positive estimate advantages over Giza 178 as commercial restorer line for anther length, anther breadth, filament length, and duration of floret opening (Appendix A).

On the contrary, some new restorer lines showed significant and highly significant positive estimate advantages over better parent as well as over Giza 178 as a commercial variety, for hulling percentage, milling percentage, and head rice percentage (Appendix A) and Figure 11.

## 4. Discussion

### 4.1. Developing New Iso-Cytoplasmic Restorers Lines 

Awad-Allah 2006 and 2011 [17,18] report that M2 is a dominant marker linked to *Rf1* gene on chromosome 1. All studied materials including parental lines (Giza 178, 86945-L, BG 33-5, and BG 34-8) have the allele of *Rf1*. Rice microsatellite RM3425 was used to detect the *Rf3* gene in the tested lines. The results show that the RM3425 marker showed inconstant data with the known restorer lines. The rice microsatellite RM 171 marker is known to be linked with the *Rf4* gene on chromosome 10 in WA CMS lines. The data revealed a band detected by RM 171 marker in studied genotypes, which suggests that these lines may have the allele of the *Rf4* gene.

Based on the results of molecular analysis and field analysis from Awad-Allah 2006 and 2011 [17,18], the selection in the progenies of the promising hybrids IR79156A/86945-L, G46A/Giza 178, G46A/BG 33-5, G46A/86945-L, G46A/BG 34-8, and IR69625A/BG 34-8 to grow for producing F_2_ started in 2013 and went from F_2_ up to F_7_ in 2019. Promising lines were selected and grown along with the parental lines and evaluated for phenotypic performance and yielding ability during 2019 and 2020 rice seasons. The newly developed restorer lines contain restorer genes from the parents. Grain yield is the most important trait for comparing the performance of restorer lines as it reflects the performance of all component traits. Most restorer lines (21 lines) were found to possess a grain yield higher than the check variety. As iso-cytoplasmic restorer lines are derived from rice hybrids by continued selfing and selection, they all possess the sterile cytoplasm from the female line (CMS). Wild abortive cytoplasm (WA) and Gambica cytoplasm have a significant effect on panicle excretion; it is observed that lines resulting from CMS as female parents show incomplete panicle excretion [41]. Thus, variation in the range of panicle excretion is observed in each generation, and by exercising appropriate selection pressure on the level of panicle excretion in segregating generations, there is an opportunity for rectifying and improving panicle excretion in the hybrids as well as in restorer lines [24]. Spikelet fertility is another important trait that would help in harvesting better yield.

Iso-cytoplasmic restorers, based on their performances, were identified. The identified lines can be screened for the presence of fertility genes and their fertility restoration behavior by crossing with CMS lines [23]. The authors of [24] report that, based on the association among different traits observed, simultaneous selection for the number of tillers, panicle length, and spikelet fertility would enable the improvement of yield per plant. They also observe that in regression analysis, the contribution of the number of effective tillers per plant, followed by panicle length and grain weight, to an explanation of yield per plant was high and significant. Hence, selection for these characters will help enhance yield as they are mutually and directly associated with grain yield. Traits showing higher variability can provide higher genetic gain in breeding programs [42] and have been used in rice for subdividing observed variation and studying interrelationships among different traits.

This method was efficient in developing a promising is cytoplasmic restorer lines from a highly diverse set of 390 iso-cytoplasmic restorers derived from diverse rice hybrids, ensuring coverage of phenotypic traits without the comparison of relative characteristics within the genotypes. The core set of iso-cytoplasmic restorers can be used for further improvement of restorers while the promising iso-cytoplasmic restorer lines can help in developing heterotic hybrids when crossed with diverse non-parental CMS lines [24].

### 4.2. Mean Performance

For effective restoration, restorers having synchronized flowering duration as male sterile lines are required to obtain a higher seed yield, because this depends on the amount of pollen supplied from the male parent during the flowering period [43]. Thus, lines with ~81 days for G46A and ~100 days for other cytoplasmic male sterile lines CMS to 50% flowering are most desirable. To achieve optimal flowering synchronization for good out-crossing, the CMS line should bloom 1–2 days before the restoration line [44]. All selected restorer lines under this study were found to be desirable in this regard. Concerning plant height, restorer lines with plant height greater than the male sterile lines have desirable plant height. In hybrid rice seed production, the plant height of parental lines plays a significant role in determining the extent of the seed set [45]. In this study, the five newly developed restorer lines, NRL 70, NRL 73, NRL 72, NRL 68, and NRL 71, were short in plant height and showed the lowest mean values (Appendix A); these lines may be useful in as inbred varieties and the improvement of a new maintainer line in a hybrid rice breeding program. However, the rice genotypes NRL 55, NRL 58, NRL 59, NRL 54-2, and NRL 65 showed the highest values of plant height in the first season and the second season; these rice genotypes were desirable for their good restorer lines. As the plant heights of popular male sterile lines, namely IR 69625A and IR 79156A, are less than 111 cm, restorer lines of plant height around 115 to 125 cm are highly desirable. In the case of the male sterile line G46A, it showed a plant height less than 92 cm, where restorer lines of plant height around 100 to 115 cm are highly desirable. The line derived from IR69625A/BG 34-8 and G46A/BG 34-8 was found to possess the desired plant height. The number of panicles per plant is believed to be closely associated with high grain yield per plant. Thus, the genotypes with a higher number of panicles per plant were identified [46]. The length of the panicle is closely related to the number of spikelets in the panicle, and then with the grain yield. An increase in the length of the panicle often leads to an increase in the grain yield. The number of spikelets per panicle and spikelet fertility are closely related to the grain yield per plant, and it is one of the most important characteristics associated with the grain yield pre-plant. The weight of a thousand grains is one of the important traits that correlate and affect the grain yield/plant, and it is an important trait in the breeding of hybrid rice in Egypt, where the high grain weight is desirable.

Concerning grain yield and agronomic traits, the genotypes NRL 59, NRL 55, NRL 62, NRL 63, and NRL 66 gave high grain yields and desirable values of other traits (Appendix A). Thus, some of these genotypes can be used as a source for developing new hybrid combinations in rice breeding programs. The authors of [38] report that the choice of the parents is a crucial step in breeding programs for improving new lines. Floral morphologies are important, decisive factors in mating systems in rice [27]. The highest mean values of floral traits studied, i.e., anther length (mm), anther breadth (mm), filament length (mm), and duration of floret opening (min), of all the studied genotypes were observed in the genotype NRL 54-2 (Appendix A). 

The anther filament starts to elongate rapidly after the opening of the spikelet, which increases the possibility of outcrossing in the fields of hybrid rice seed production [47]. An increase in the length of filament thread leads to an increase in the opportunity for pollen to spread, which leads to an improvement in the cross-pollination rate (outcrossing) and thus an increase in the seed set in seed production of hybrid rice. A major factor in the increase in the rate of outcrossing in parental lines is the duration of flowering in flowers [27]. Anther length, anther breadth, and filament elongation are the most important floral traits for pollen parents in hybrid rice seed production fields [47]. The floral trait of parental lines especially restorer lines is the main factor in hybrid rice seed production because it influences the outcrossing between parental lines [25].

With regard to grain quality traits, the genotypes NRL 52, NRL 59, NRL 55, and NRL 46 were recorded with one or more desirable traits (Appendix A). Therefore, we can use these genotypes for improving some new hybrids with good grain quality traits; the mean performance of these rice genotypes indicated promise for the yield performance associated with a high number of panicles per plant, number of spikelets per panicle, panicle length, number of filled grains/panicle, spikelets fertility percentage, and 1000-grain weight.

Giza 178 is moderately tolerant to the stem borer insect at its ripening phase. While the newly developed restorer lines showed tolerance to the insect at the ripening phase, in general the striped stem borer causes significant losses for susceptible genotypes, along with blast (Appendix A).

### 4.3. Test Cross Experiment

A test cross nursery to identify restorer and maintainer in hybrid rice breeding is the first step for developing new hybrids. Pollen fertility and spikelet fertility analysis is used to identify restorer and maintainer lines [48]. Pollen fertility and spikelet fertility are useful for studying the genetics of the restorer gene [49]. The pollen fertility percentage of tested hybrids was higher than 85% (Appendix A). Similar results were obtained by [48,50,51,52,53], while the spikelet fertility percentage of tested hybrids was higher than 85% (Appendix A). The 22 promising lines were identified as effective restorers for two CMS lines; IR69625A and Gang46A. The high fertility of these hybrid combinations can be useful in developing new, promising hybrid combinations. The previous results agree with the results by [24,48,50,51,52].

### 4.4. Analysis of Variance 

Analysis of variance showed highly significant differences between all genotypes for all the studied traits (Table 2), which indicates that each genotype is genetically different from the other for yield, its components, floral traits, and grain quality, indicating that there is variability among the studied lines and they would respond positively to selection. The set of genotypes used in the current study indicated that there were statistically significant differences between them among all the traits that were studied, and these results were consistent with the findings of [54,55,56,57,58]. 

### 4.5. Phenotypic, Genotypic Coefficient of Variation and Genetic Advance

Genetic variability is a prerequisite for the selection of superior genotypes over the existing cultivars. Estimates of PCV%, GCV%, and genetic advances as revealed from the results indicate that there is a large amount of variability among the genotypes of all the traits studied. In general, the phenotypic coefficient of variation was higher than the genotypic coefficient of variation suggesting the influence of environment on the expression of these traits. A similar observation has also been noted by [59,60]. 

The results of this study revealed that high values, moderate values, and lower values were estimated for phenotypic and genotypic coefficient of variation for the studied traits, but with a narrow difference between phenotypic and genotypic coefficients of variation for all studied traits (Table 3). This suggests a limited role for environmental variance in the expression of these traits. Thus selection based on the genetic performance of the characters would be effective to bring about a significant improvement in these traits. Selection in breeding programs is based on measurements of phenotypic characteristics and genotypic variability and is measured through analysis of variance; similar observations have been made by [60,61]. A similar observation has also been noted by [56,57,62], who reported that high genetic advance and genotypic coefficient of variation were observed for most of these characteristics.

The genetic advance is a useful indicator of the progress that can be expected as a result of selection in the related population. Heritability in conjunction with genetic advances would give a more reliable index of selection value [37]. High PCV and GCV values and high genetic advances were recorded for these traits, suggesting further improvement of lines for these characteristics for further selection and subsequent use in the breeding program. Similar results were also reported by [54,55,56,57,58,61,63]. In general, heritability along with genetic advances can be useful in the selection program. The genetic advance in percentage (expected) of mean was high for plant height, number of panicles per plant, number of spikelets per panicle, number of filled grains per panicle, panicle weight, grain yield per plant, anther length, anther breadth, duration of floret opening, and head rice percentage (Table 3). Moreover, the genetic advance in percentage (expected) of mean was moderate for days to 50% of heading, panicle length, 1000-grain weight (g), and filament length, while low genetic advances in percentage (expected) of mean were observed for spikelet fertility percentage, hulling percentage, and milling percentage. Similar results were observed by [54,55,58,61,63,64].

High heritability with high genetic advance as a percentage of the mean indicates that these characters are largely controlled by additive gene action, which indicates that improvement in these characters is possible through mass selection and progeny selection. High heritability with low genetic advance was observed for spikelet fertility percentage, hulling percentage, and milling percentage. This suggests non-additive gene action for the expressions of these traits. The high heritability was exhibited due to the favorable influence of environment rather than genotype and selection for such traits might not be rewarding.

### 4.6. Estimation of Heritability

The estimates of heritability as a predictor are a tool in expressing the reliability of phenotypic value. Therefore, high heritability helps in effective selection for a particular trait. In this study, the traits expressed moderate to high heritability estimates ranging from 66.19 to 99.41% and 48.84 to 99.40% in the first and second years, respectively (Table 3). High heritability was observed for the traits days to 50% heading, plant height, panicle length, number of spikelets per panicle, number of filled grains per panicle, panicle weight, 1000-grain weight, grain yield per plant, anther length, anther breadth, filament length, duration of floret opening, and head rice percentage. Similar results were reported by [65] for days to 50% heading, plant height, panicle length, number of filled grains per panicle, 1000-grain weight, and [66] for days to 50% heading, number of filled grains per panicle, 1000-grain weight, and grain yield per plant. See [54,55,58,63] for anther length, anther breadth, filament length, duration and of floret opening. 

Moderate heritability was observed in the number of panicles per plant, spikelet fertility percentage, hulling percentage, and milling percentage in the second year (Table 3), Similar results were observed by [64] for all studied traits. 

High heritability values indicate that the traits under study are less affected by the environment in their expression. Therefore, a plant breeder may safely select them based on the phenotypic expression of these traits in the individual plant by adopting simple selection methods.

### 4.7. The Advantage over Better Parent and Commercial Variety

The data showed that the percentage of advantage over better parent and Giza 178 as the commercial variety was significant and highly significant among the genotypes for all the traits studied in the two years, indicating that the selection is effective in the genetic improvements for these traits (Appendix A). For days to 50% heading and plant height, many lines showed significant and highly significant negative estimates over better parent as well over Giza 178 as a commercial variety; this is useful in using these lines as inbred variety. Additionally, 15 new lines showed significant and highly significant positive estimates values over Giza 178 as a commercial variety for plant height, this is useful in using these lines as restorer lines. Regarding the number of panicles/plant, 13 newly developed restorer lines showed significantly and highly estimated values in both years. Genotypes with more panicles per plant had a higher grain yield per plant [46,50]. Concerning panicle length, 13 new restorer lines gave significant or highly estimated values in the first and second years. The genotype with a longer panicle length is desirable since the lengthy panicles are generally associated with a higher number of spikelets per panicle resulting in higher grain yield; similar results were obtained by [67]. For the number of spikelets per panicle, 12 newly developed restorer lines showed significant and highly estimated values in the first and second year. The number of spikelets per panicle is one of the main yield components that improve grain yield [67]. Previous data agree with this [68]. The number of filled grains/panicles is one of the most important traits that directly influence grain yield potentiality in rice varieties and hybrids. The results revealed that 12 lines showed significant and highly significant positive estimates over Giza 178 as a commercial restorer line for filled grains/panicles. For panicle weight, the newly developed restorer lines showed significant or highly significant positive estimates over Giza 178 as commercial restorer line. An increase in the value of the weight of the seed leads to an increase in the yield. Concerning 1000-grain weight, most of the lines showed significant and highly significant positive estimates over better parent as well over Giza 178 as a commercial variety; this is useful in using these lines as restorer lines to produce new promising hybrids of rice and desirable for farmers in Egypt. Regarding grain yield, 19 lines showed significant and highly significant positive estimates over better parent as well over Giza 178 as a commercial variety; this is useful in using these lines as inbred variety and restorer lines to produce new promising hybrids of rice and is desirable for farmers in Egypt (Appendix A).

Moreover, most of the newly developed restorer lines showed significant and highly significant positive estimates over Giza 178 as a commercial restorer line for anther length, anther breadth (Appendix A), filament length, and duration of floret opening. Anther length, anther breadth, and filament elongation are the most important floral traits for pollen parents in hybrid rice seed production fields [47]. The floral traits of parental lines, especially restorer lines, are the main factors in hybrid rice seed production because they influence outcrossing between parental lines [25]. A major factor in the increase in the rate of outcrossing in parental lines is the duration of floret opening [27].

## 5. Conclusions

The presence of genetic variability is a basic requirement in any crop improvement program. In conclusion, there are significant differences between the genotypes of all the traits under study. This indicates that there is broad scope for the selection of promising genotypes from the present set of genotypes for yield, components, floral traits, and some grain quality traits improvement. The new restorer lines NRL 59, NRL 55, NRL 62, NRL 63, NRL 66, and NRL 54-2 were promising for the grain yield and other traits. Thus, some of these genotypes can be used as a new rice variety plus a source for developing new hybrid combinations in rice breeding programs. The magnitude of the phenotypic coefficient of variation was higher than the genotypic coefficient of variation for all the traits under study. Both GCV% and PCV% were high for spikelets per panicle, the number of filled grains per panicle, and panicle weight. The genetic advance in the percentage of mean was high for plant height, number of panicle per plant, panicle length, number of spikelets per panicle, number of filled grains per panicle, panicle weight, grain yield per plant, anther length, anther breadth, duration of floret opening, and head rice percentage. The percentage of advantage over better parent and the Giza 178 commercial variety was significant and highly significant, and desirable values were found among the genotypes for all the traits studied in the two years, indicating that the selection is effective in the genetic improvements for these traits. The 22 promising restorer lines with good floral traits were identified as effective restorers for two CMS lines, IR69625A and Gang46A. The high fertility of these hybrids can be usefully used in developing promising hybrid combinations.

## Figures and Tables

**Figure 1 genes-13-00458-f001:**
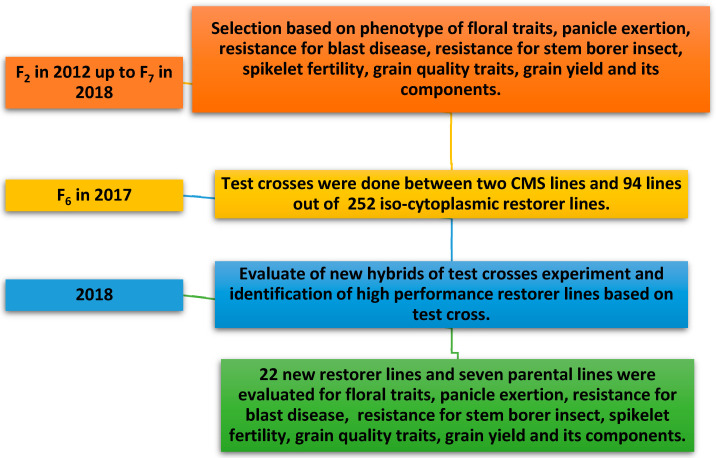
Breeding scheme illustrating the development of 22 promising new iso-cytoplasmic restorer lines in rice through the Rice Breeding Program, Rice Research Section, Field Crops Research Institute, Agricultural Research Center, Giza, Egypt.

**Figure 2 genes-13-00458-f002:**
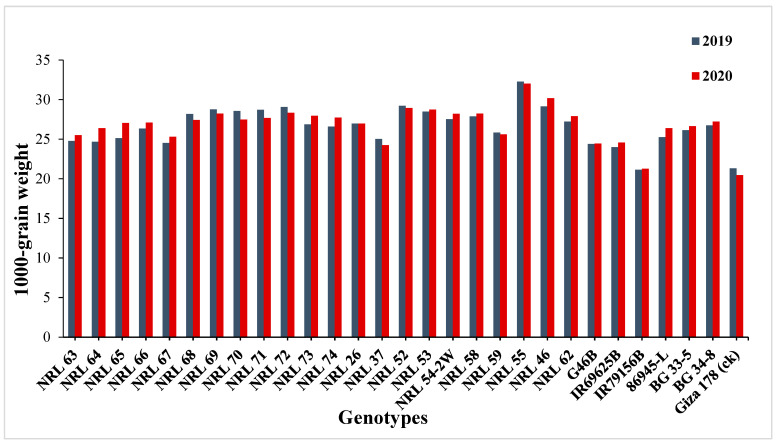
Mean performances for 1000 grain weight (g) of the studied genotypes during 2019 and 2020 growing season.

**Figure 3 genes-13-00458-f003:**
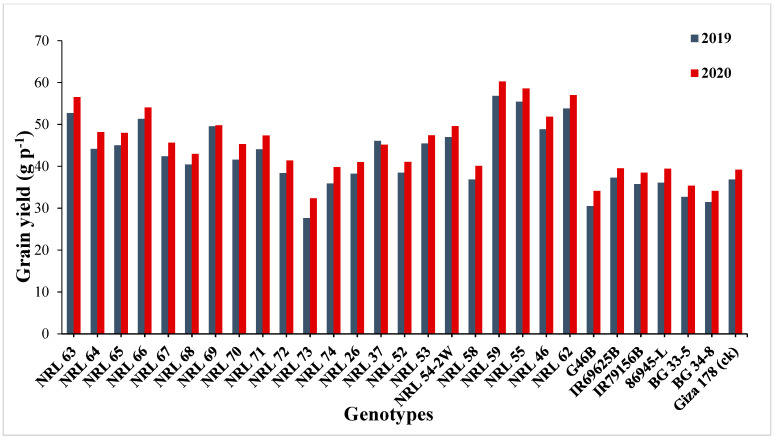
Mean performances for grain yield per plant (g) of the studied genotypes during 2019 and 2020 growing season.

**Figure 4 genes-13-00458-f004:**
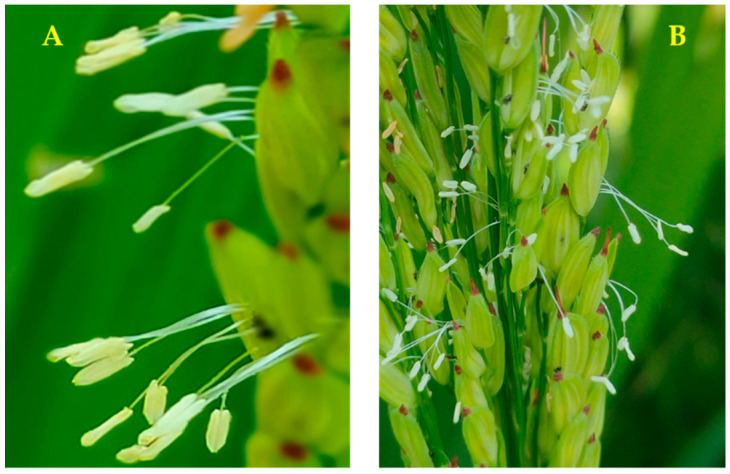
(**A**) picture of the flowers of one of the newly developed lines showing the flowering and (**B**) picture of the flowers of one of the newly developed breeds, showing the length of anthers, filament, and size of the anthers.

**Figure 5 genes-13-00458-f005:**
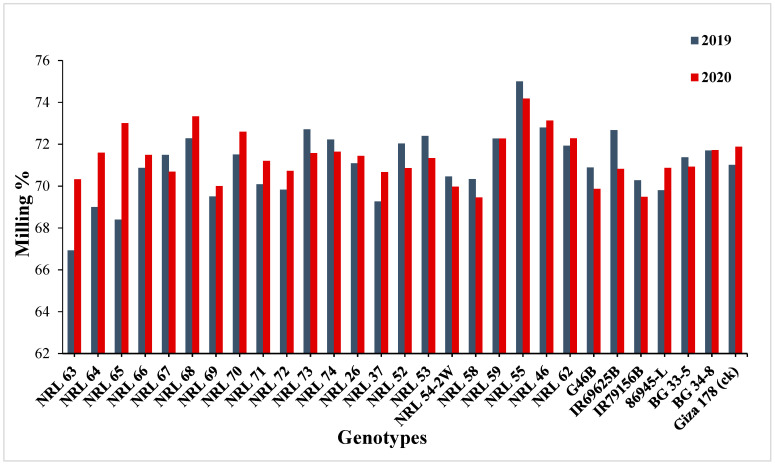
Mean performances for milling percentage of the studied genotypes during the 2019 and 2020 growing seasons.

**Figure 6 genes-13-00458-f006:**
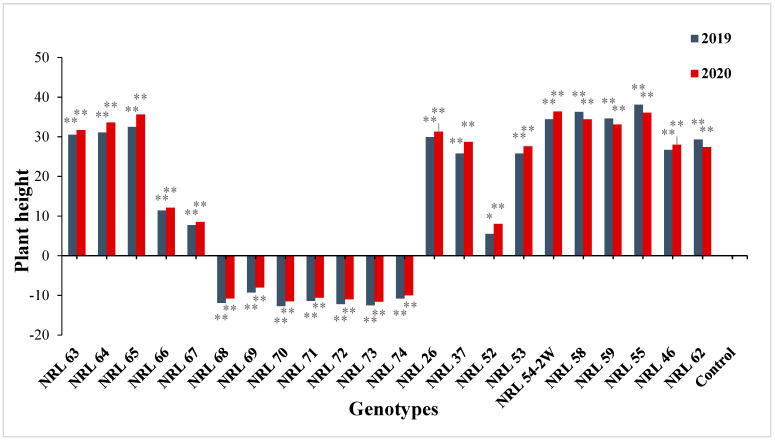
The percentage of advantage over commercial variety for plant height of studied genotypes during the 2019 and 2020 growing seasons. Means **: Highly significant at 1%; *: Significant at 5%.

**Figure 7 genes-13-00458-f007:**
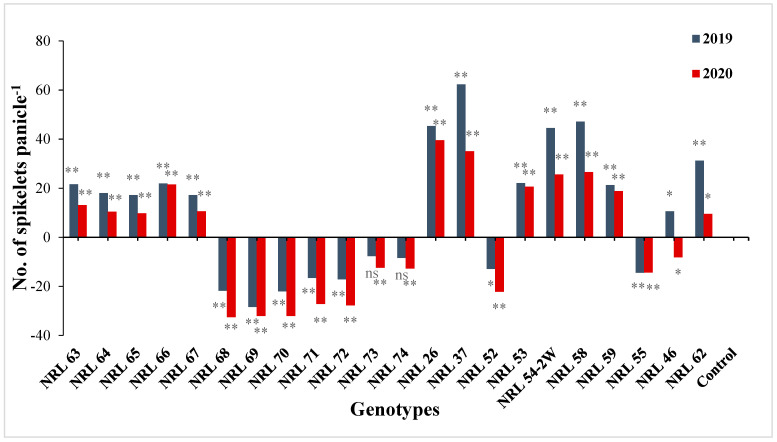
The percentage of advantage over commercial variety for number of spikelets per panicle of studied genotypes during the 2019 and 2020 growing seasons. Means **: Highly significant at 1%; *: Significant at 5%; ns: Non-significant.

**Figure 8 genes-13-00458-f008:**
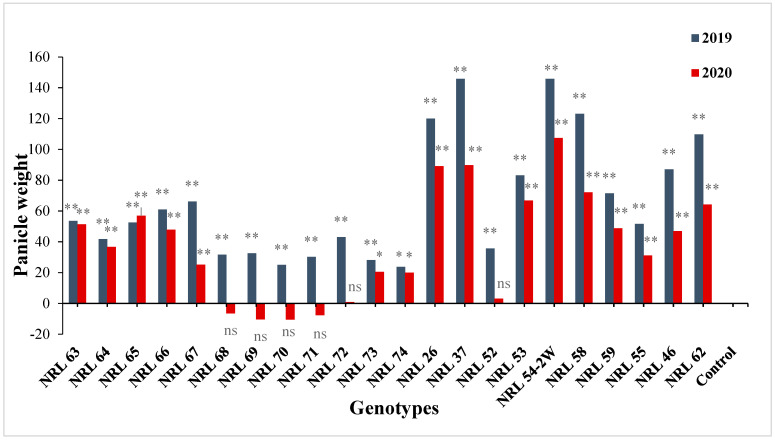
The percentage of advantage over commercial variety for panicle weight of studied genotypes during the 2019 and 2020 growing seasons. Means **: Highly significant at 1%; *: Significant at 5%; ns: Non-significant.

**Figure 9 genes-13-00458-f009:**
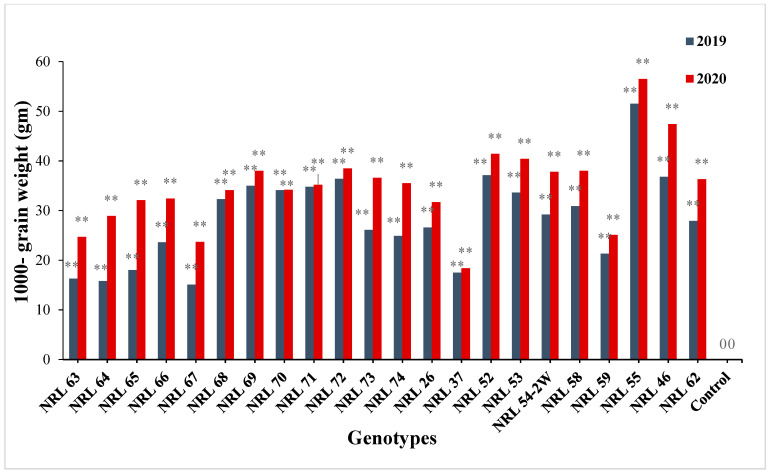
The percentage of advantage over commercial variety for 1000-grain weight of studied genotypes during the 2019 and 2020 growing seasons. Means **: Highly significant at 1%.

**Figure 10 genes-13-00458-f010:**
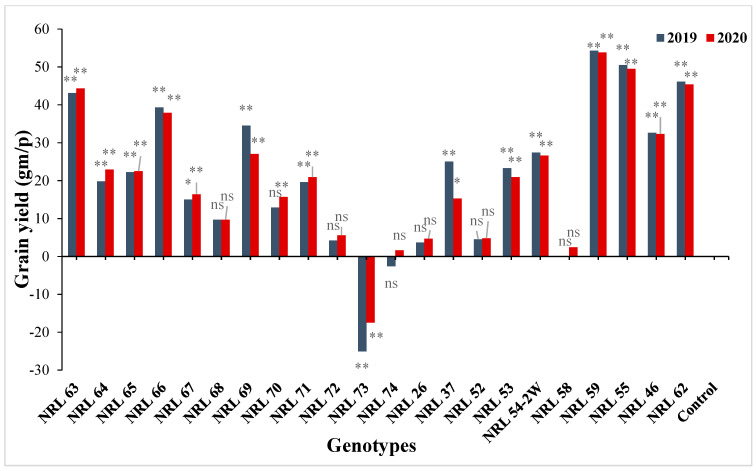
The percentage of advantage over commercial variety for grain yield/plant of studied genotypes during the 2019 and 2020 growing seasons. Means **: Highly significant at 1%; *: Significant at 5%; ns: Non-significant.

**Figure 11 genes-13-00458-f011:**
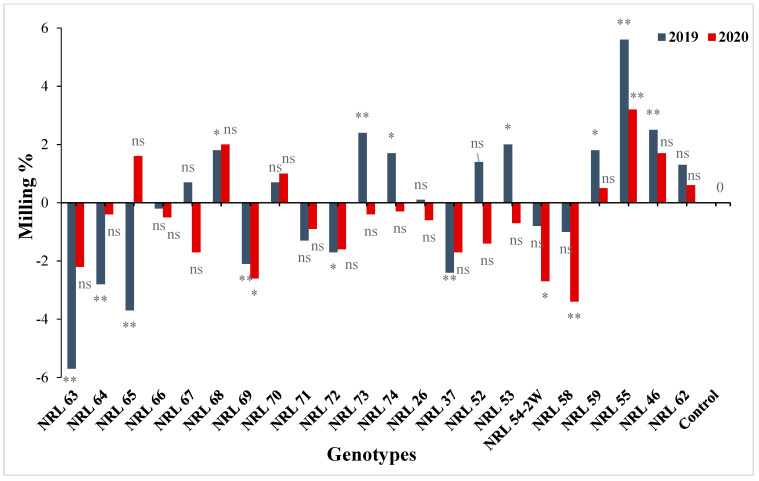
The percentage of advantage over commercial variety for milling percentage of studied genotypes during the 2019 and 2020 growing seasons. Means **: Highly significant at 1%; *: Significant at 5%; ns: Non-significant.

**Table 1 genes-13-00458-t001:** Names and parentage of the selected genotypes under study.

No.	Lines	Parentage	Description
1	NRL 63	IR79156A/86945-L	New developed restorer
2	NRL 64	IR79156A/86945-L	New developed restorer
3	NRL 65	IR79156A/86945-L	New developed restorer
4	NRL 66	G46A/Giza 178	New developed restorer
5	NRL 67	G46A/Giza 178	New developed restorer
6	NRL 68	G46A/BG 33-5	New developed restorer
7	NRL 69	G46A/BG 33-5	New developed restorer
8	NRL 70	G46A/BG 33-5	New developed restorer
9	NRL 71	G46A/BG 33-5	New developed restorer
10	NRL 72	G46A/BG 33-5	New developed restorer
11	NRL 73	G46A/BG 33-5	New developed restorer
12	NRL 74	G46A/BG 33-5	New developed restorer
13	NRL 26	G46A/BG 33-5	New developed restorer
14	NRL 37	G46A/BG 33-5	New developed restorer
15	NRL 52	G46A/86945-L	New developed restorer
16	NRL 53	G46A/86945-L	New developed restorer
17	NRL 54-2	G46A/BG 34-8	New developed restorer
18	NRL 58	G46A/BG 34-8	New developed restorer
19	NRL 59	G46A/BG 34-8	New developed restorer
20	NRL 55	IR69625A/BG 34-8	New developed restorer
21	NRL 46	IR69625A/BG 34-8	New developed restorer
22	NRL 62	IR69625A/BG 34-8	New developed restorer
23	G46A, B	Erjiu’ai 7/V41B//Zhenshan 97/Ya’aizao	cytoplasmic male sterile and its maintainer line
24	IR69625A, B	not available	cytoplasmic male sterile and its maintainer line
25	IR79156A, B	not available	cytoplasmic male sterile and its maintainer line
26	86945-L	not available	Restorer line ^1^
27	BG 33-5	not available	Restorer line ^1^, drought tolerant high grain quality
28	BG 34-8	not available	Restorer line ^1^, drought tolerant high grain quality
29	Giza 178 (ck)	Giza 175/Milyang 49	Commercial Restorer line,

NRL: New Restorer line, ^1^: Identified by Awad-Allah 2011 [17].

**Table 2 genes-13-00458-t002:** Analysis of variance for the grain yield and the contributing traits of the studied genotypes during the 2019 and 2020 growing season.

	Mean Sum of Squares						CV	
	2019			2020				
	Rep.	Genotype	Error	Rep	Genotype	Error	2019	2020
d.f	2	28	56	2	28	56		
days to 50% of heading (day)	0.10	248.64 **	0.49	0.87	246.99 **	0.50	0.736	0.737
plant height (cm)	7.99	820.95 **	6.53	13.35	805.15 **	10.33	2.52	3.05
number of panicles /plant	1.83	19.30 **	2.81	2.21	12.44 **	2.58	10.05	9.47
panicle length (cm)	0.09	16.60 **	0.55	1.12	17.60 **	0.67	3.22	3.50
number of spikelets /panicle	193.31	3979.85 **	77.79	109.21	4078.95 **	57.38	5.20	4.47
spikelet fertility percentage	5.05	19.09 **	2.87	1.57	12.01 **	1.25	1.85	1.2
number of filled grains per panicle	81.26	3290.61 **	73.46	103.89	3442.09 **	49.37	5.53	4.47
panicle weight (g)	0.02	3.50 **	0.11	0.11	3.85 **	0.14	7.58	8.39
1000-grain weight	0.20	17.34 **	0.24	0.08	16.51 **	0.18	1.84	1.57
grain yield/plant	11.30	180.31 **	10.46	0.47	173.85 **	8.37	7.69	6.44
Anther length (mm)	0.08	0.51 **	0.04	0.07	0.52 **	0.04	9.05	9.13
Anther breadth (mm)	0.0003	0.015 **	0.0004	0.0001	0.014 **	0.0003	4.59	4.1
Filament length (mm)	0.34	1.24 **	0.16	0.37	1.44 **	0.16	5.98	6.04
Duration of floret opening (min)	15.998	1292.18 **	12.67	12.09	1291.53 **	9.71	2.74	2.41
Hulling (%)	0.88	3.92 **	0.47	0.58	3.28 **	0.62	0.85	0.98
Milling (%)	0.91	7.73 **	0.50	0.73	4.08 **	1.06	1.00	1.44
Head rice %	3.98	262.29 **	2.68	25.79	251.13 **	9.11	2.82	5.17

**: Highly significant at 1%.

**Table 3 genes-13-00458-t003:** Estimates of variability parameters for some agronomic and grain quality traits in rice lines over two years.

	GCV%		PCV%		GA		Heritability %		GA%	
	2019	2020	2019	2020	2019	2020	2019	2020	2019	2020
days to 50% of heading (day)	9.54	9.48	9.57	9.51	18.68	18.62	99.41	99.40	19.60	19.47
plant height (cm)	15.79	15.51	15.98	15.81	33.54	32.90	97.65	96.25	32.15	31.35
number of panicles per plant	14.06	10.70	17.28	14.29	3.93	2.80	66.19	10.70	23.56	16.51
panicle length (cm)	10.03	10.13	10.54	10.71	4.54	4.62	90.64	89.33	19.67	19.71
number of spikelets per panicle	21.27	21.62	21.90	22.08	72.17	73.86	94.36	95.90	42.57	43.62
spikelet fertility percentage	2.54	2.04	3.15	2.37	3.87	3.36	65.29	74.22	4.23	3.62
number of filled grains per panicle	21.14	21.41	21.85	21.87	65.26	67.81	93.59	95.82	42.13	43.16
panicle weight (g)	23.84	24.53	25.02	25.92	2.09	2.17	90.82	89.53	46.81	47.80
1000-grain weight (g)	8.98	8.70	9.17	8.84	4.82	4.73	95.98	96.85	18.13	17.63
grain yield/plant	17.88	16.53	19.46	17.74	14.24	14.26	84.41	86.82	33.84	31.73
Anther length (mm)	17.61	17.92	19.80	20.11	0.73	0.73	79.13	79.37	32.27	32.88
Anther breadth (mm)	15.59	15.05	16.18	15.59	0.14	0.14	92.84	93.1	30.93	29.9
Filament length (mm)	9.02	9.87	10.83	11.57	1.03	1.15	69.46	72.74	15.49	17.34
Duration of floret opening (min)	15.87	15.95	16.10	16.13	41.92	42.11	97.11	97.78	32.21	32.49
Hulling (%)	1.340	1.17	1.59	1.53	1.87	1.48	71.18	58.58	2.33	1.84
Milling (%)	2.19	1.41	2.40	2.01	2.91	17.54	82.68	48.84	4.09	2.03
Head rice (%)	16.01	15.39	16.25	16.24	18.87	17.54	97.00	89.85	32.47	30.05

Note: GCV% = Genotypic coefficient of variation, PCV% = Phenotypic coefficient of variation, GA = Genetic advance, GA (%) = Genetic advance as percent of mean.

## Data Availability

Relevant data applicable to this research are within the paper.

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
