# Peer review of "Development of New Restorer Lines Carrying Some Restoring Fertility Genes with Flowering, Yield and Grains Quality Characteristics in Rice (Oryza sativa L.)"

_genes, 2022, doi:10.3390/genes13030458_

Round 1
Reviewer 1 Report
- In the introduction, line 59, it is recommended authors check the meaning of ‘heterotrophic’ and revise it in the manuscript.
- In the introduction, line 75, what about the full name of “WA”?
- In the materials and methods, line 130, why use Giza 178 as a check? what about its agronomic performance? the parents used in the cross combinations need to describe detail in the manuscript.
- Line 161, Table (1), it is recommended authors remove brackets.
- In figure 1, the authors describe using the restorer lines to evaluate the resistance for the bests or insects, but the data is not shown in the manuscript. It is recommended authors can describe this information in the manuscript.
- In Tables 1, and 4-9, It is recommended authors need to add the note to specify what the asterisks represent? and what is the ns?
- In Tables 3-9, the decimal point of the numbers in the table needs to be aligned.
- In the results, line 409, the punctuation mark must be corrected.
- In the results, I cannot find the description the Tables 8 and 9. It is recommended authors need to revise.
- In Tables 8 and 9, the traits crosses, what is G1-22? it is recommended authors add descriptions in the ‘methods and materials' or have a note in the table.
- In the discussion, lines 511 and 598, the numbers 4.3 and 4.2 need to exchange each other.
Author Response
Attached file the response to reviewer one

Reviewer 2 Report
This paper reports fertility restorer lines of rice, which is important for hybrid seed production. However, the paper needs several changes as given below.
1. Title of the paper must be changed to "New Iso-Cytoplasmic Restorer Lines of Rice (Oryza Sativa L.) with Improved Floral, Grain Yield, Yield Components, And Grain Quality Traits"
2. Please check the affiliations. Affiliation of 1st and 2nd authors must be the same. Check the affiliations of all authors and number them correctly.
3. Introduction: Include the past examples of restorer lines in rice and how your lines are different from the other lines? and what is known Rf genes linked with this trait?
4. Introduction: Include the known CMS genes in this part.
5. Please improve the grammar of the whole manuscript as it seems a weak link.
6. In the whole document, there is a lot of data for different estimates, but there is no data of lines for yield and quality traits. Please provide graphs that show the performance of these lines over years. Also, provide the performance of negative and positive controls for comparison.
7. If possible, show the diversity of relevant gene/s in your lines through PCR or qPCR.
Author Response
Attached file the response to reviewer two

Round 2
Reviewer 1 Report
None
Author Response
The reviewer one write "none" Comments and Suggestions for Authors.
Reviewer 2 Report
Dear authors,
Please provide original data for 1000 grain weight, grain yield, and milling percentage instead of given percentage advantage in Figures 1, 2, and 3.
Author Response
attached file the response to reviewer two.
